# Comparative Analysis of PGRP Family in Polymorphic Worker Castes of *Solenopsis invicta*

**DOI:** 10.3390/ijms252212289

**Published:** 2024-11-15

**Authors:** Zhanpeng Zhu, Hongxin Wu, Liangjie Lin, Ao Li, Zehong Kang, Jie Zhang, Fengliang Jin, Xiaoxia Xu

**Affiliations:** State Key Laboratory of Green Pesticide, “Belt and Road” Technology Industry and Innovation Institute for Green and Biological Control of Agricultural Pests, College of Plant Protection, South China Agricultural University, Guangzhou 510642, China; zhuzhanpeng@stu.scau.edu.cn (Z.Z.); scauwhx@stu.scau.edu.cn (H.W.); liangjie@stu.scau.edu.cn (L.L.); 20233138166@stu.scau.edu.cn (A.L.); kzh980817@163.com (Z.K.); 13560437669@163.com (J.Z.)

**Keywords:** *Solenopsis invicta*, *Metarhizium anisopliae*, PGRP family, immune response

## Abstract

Peptidoglycan recognition proteins (PGRPs) are a class of pattern recognition receptors (PRRs) that activate the innate immune system in response to microbial infection by detection of peptidoglycan, a distinct component of bacterial cell walls. Bioinformatic studies have revealed four PGRPs in the red imported fire ant *Solenopsis invicta*; nonetheless, the mechanism of the immune response of *S. invicta* induced by pathogens is still poorly understood. The peptidoglycan recognition protein full-length cDNA (designated as *SiPGRP*-*S1/S2/S3/L*) from *S. invicta* was used in this investigation. According to the sequencing analysis, there was a significant degree of homology between the anticipated amino acid sequence of SiPGRPs and other members of the PGRPs superfamily. Molecular docking studies demonstrated that SiPGRPs show strong binding affinity for a variety of PGN substrates. Additionally, tissue distribution analysis indicated that *SiPGRPs* are primarily expressed in several tissues of naïve larvae, including fat body, hemocytes, head, and thorax, as detected by quantitative real-time PCR (RT-qPCR). Microbial challenges resulted in variable changes in mRNA levels across different tissues. Furthermore, the antibacterial effects of antimicrobial peptides (AMPs) produced by major ants infected with *Metarhizium anisopliae* were assessed. These AMPs demonstrated inhibitory effects against *M. anisopliae*, *Staphylococcus aureus*, and *Escherichia coli*, with the most pronounced effect observed against *E. coli*. In conclusion, SiPGRPs act as pattern recognition receptors (PRRs) that identify pathogens and initiate the expression of AMPs in *S. invicta*, this mechanism contributes to the development of biopesticides designed for the targeted control of invasive agricultural pests.

## 1. Introduction

A diverse array of natural pathogens, including bacteria, fungi, and viruses, invade insects [1]. Insects defend against these invading pathogens through innate immunity, which encompasses both humoral and cellular immunity [2]. Pattern recognition receptors (PRRs) play a crucial role in recognizing pathogen-associated molecular patterns (PAMPs) and initiating various immune responses in insects [3]. Among these, peptidoglycan recognition proteins (PGRPs) are significant PRRs found across a wide range of organisms, from insects to mammals [4].

Insects primarily rely on two distinct signaling pathways, Toll and immune deficiency (IMD), to induce the expression of antimicrobial peptides (AMPs) when confronted with pathogens in the context of humoral immunity [5]. Peptidoglycan (PGN) is a specific polymer present in the cell walls of most bacteria, with notable differences between Gram-positive and Gram-negative bacteria. In Gram-negative bacteria, the lysine at the third position of the peptide chain is substituted with meso-diaminopimelic acid (DAP). DAP-type PGNs can be recognized by certain PGRPs, thereby activating the IMD pathway [6]. LYS-type PGNs are predominant in Gram-positive bacteria and activate the Toll pathway similarity fungi [7].

PGRPs have been extensively studied in various insects. The structural domain of PGRPs is highly conserved and exhibited significant homolog to T7 lysozyme [8]. They disrupt amide bonds in sugar chains through a mechanism akin to N-acetylmuramoyl-L-alanine amidase [9]. Based on their amidase activity, PGRPs are categorized into catalytic PGRPs and recognition PGRPs [10]. The first PGRP identified in insects was found in the hemolymph of *Bombyx mori* [11]. Currently, 13 and 12 PGRPs have been characterized in the genomes of *Drosophila melanogaster* and *B. mori*, respectively [12,13]. In *D. melanogaster*, PGRP-SA, PGRP-SD, PGRP-LA, PGRP-LC, PGRP-LD, PGRP-LE, and PGRP-LF lack amidase activity but retain the ability to bind and recognize PGN. Notably, PGRP-SA and PGRP-SD can synergize with Gram-negative binding protein (GNBP) to activate the Toll pathway [7,14]. GNBP3 is crucial in the response to fungal infestation [15]. The depletion of PGRP-SC1 and PGRP-SC2 results in the over-activation of the IMD pathway [16]. PGRP-LC and PGRP-LE function as upstream pattern recognition receptors (PRRs) of the IMD pathway, responding to infection caused by Gram-negative bacteria [17]. Additionally, PGRP-LB specifically degrades peptidoglycan and downregulates the IMD pathway [18]. These regulators play a vital role in maintaining the homeostasis of the insect microbiome.

The red imported fire ant (RIFA), *Solenopsis invicta* (Hymenoptera: Formicidae), is recognized as one of the most significant invasive alien species [19]. This species is highly aggressive and poses a threat to human health by attacking individuals [20]. Additionally, it exhibits strong competitive abilities in new environments, leading to damage in agriculture, habitats, and biodiversity [21,22]. *S. invicta* is particularly likely to invade areas closely associated with human activities, such as farmland [23]. Although chemical pesticides remain the primary method of controlling *S. invicta* [24,25], they are not environmentally friendly and pose risks to human health, prompting a desire for the application of environmentally safe microbial insecticides for pest control [26]. *Metarhizium anisopliae* has emerged as a promising microbial pesticide, with numerous successful applications in agricultural pest management [27,28,29]. In response to the invasion of pathogenic microorganisms, insects detect pathogens through a series of recognition proteins and activate the Toll and IMD pathways to mount a defense [30].

To enhance the effectiveness of biopesticides against pests, a comprehensive understanding of the defense mechanisms employed by pests against pathogenic microorganisms is essential. However, the immune response mechanism of *S. invicta* confronted with pathogens remains inadequately understood. In the present study, we analyzed the tissue expression profiles of *SiPGRPs* and explored the immune mechanisms of *S. invicta* in response to pathogen infestation, utilizing structural prediction and comparative analysis. The findings will offer new insights into the role of PGRPs in the immune response of *S. invicta* and contribute to the development of biopesticides aimed at the targeted control of invasive agricultural pests.

## 2. Results

### 2.1. Genome-Wide Identification of PGRP Genes in S. invicta

We identified 10 candidate *PGRP* gene members from the UNIL-Sinv_3.0 genome using the Hidden Markov Model and blastp analysis. However, four of these members were located on a scaffold, suggesting that certain regions of the genome may have been incorrectly assembled. To address this, we utilized transcriptomic data to correct and refine the assembly. Following this correction, we determined the locations of the ten candidate genes through blastp analysis. After filtering out repetitive sequences, we ultimately identified four *PGRP* gene members in *S. invicta*, which include three short *PGRPs* and one long *PGRP*. The predicted signal peptides for *SiPGRP*-*S1* and *SiPGRP*-*S3* were 20 and 18 amino acids in length, respectively, while *SiPGRP*-*S2* and *SiPGRP*-*L* do not present any signal peptides. Based on their physicochemical properties (Table 1), the *PGRP* genes encode proteins that range from 194 to 446 amino acids (aa). The maximum instability index for these four proteins is 38.05, indicating that they are predicted to be stable. Only one *PGRP* exhibited a PI (isoelectric point) less than 7. Furthermore, all *PGRP* members predicted negative values for the GRAVY (grand average of hydropathicity) index, suggesting a degree of hydrophilicity.

### 2.2. Chromosome Location and Gene Duplication

The *PGRP* genes in *S. invicta* are predominantly located on chromosomes 8 and 9. We mapped the chromosomal locations of the four *PGRP* genes (Figure 1A). *SiPGRP*-*S1* and *SiPGRP*-*L* are situated on chromosome 9. Notably, due to tandem gene duplication events, *SiPGRP*-*S2* and *SiPGRP*-*S3* are located on chromosome 8 and are clustered together. We assessed the selection pressure on these tandem duplication genes, revealing a Ka/Ks value of 0.33, which is less than 1. This suggests that they have evolved under purifying selection, indicating a conserved function in *S. invicta*.

Additionally, we mapped the gene structures of the four *PGRP* genes based on their annotations (Figure 1B). It is evident that the *PGRP* genes consist of four to five exons in *S. invicta*. Given that *SiPGRP*-*S2* and *SiPGRP*-*S3* are tandem duplication genes, their gene structures exhibit similarities. The short *PGRPs* are located on the negative strand of the chromosome, whereas the long *PGRP* is situated on the positive strand.

To investigate the evolutionary history of *PGRP* genes in *S. invicta* and other species, including *Drosophila melanogaster*, *Bombyx mori*, *Monomorium pharaonis,* and *Apis cerana*, a collinearity analysis was conducted among these five species (Figure 2). The analysis identified four tandem duplicate pairs within the *DmPGRP* gene family, while both the *BmPGRP* and *AcPGRP* gene family each exhibited one tandem duplicate pair. Furthermore, the analysis revealed that the four *PGRP* genes of *S. invicta* have collinearity relationships with the corresponding genes in other species. As members of the Hymenoptera, both *M. pharaonis* and *A. cerana* possess homologous genes corresponding to the *PGRP* genes found in *S. invicta*, encompassing all *PGRP* genes from these three species. In contrast, *B. mori* and *D. melanogaster* exhibited five and eight pairs of collinearity, respectively. This suggests that the four *PGRP* genes of *S. invicta* were present during the divergence of their common ancestor with these species, indicating a relatively conservative nature in the evolutionary process.

### 2.3. Phylogenetic Analysis of the PGRP Gene Family in S. invicta

To assess the evolutionary relationships with the PGRP gene family, we utilized the protein sequences of 104 PGRPs across 21 species to construct a phylogenetic tree (Figure 3), with *Homo sapiens* serving as an outgroup. The PGRPs from *S. invicta* were categorized into three major groups. The amino acid sequences of PGRP genes in Hymenoptera exhibit a high degree of conservation, resulting in their clustering together. SiPGRP-S1 was found to cluster with other Hymenoptera PGRPs, predominantly characterized by PGRP-SC2s. In contrast, SiPGRP-S2 and SiPGRP-S3 are closely related in the phylogenetic tree, attributed to their tandem repeats, and are part of groups dominated by PGRP-SA genes. Furthermore, the group containing SiPGRP-L is primarily composed of PGRP-LC genes.

### 2.4. Multiple Sequence Alignment and Catalytic Site

The PGRP domain is a functional domain that recognizes PGN and exhibits some catalytic activity. Additionally, the structure of the PGRP domain is similar to that of T7 lysozyme. Results from multiple sequence alignments revealed that SiPGRP-S1 contains five key conserved residues associated with Zn^2+^ binding ability and amidase activity (H68, Y93, H167, T173, and C175). Two completely conserved cysteine residues (C57 and C63) form a disulfide bond that maintains structural stability. In contrast, SiPGRP-S2 and SiPGRP-S3 lacked three conserved residues, while SiPGRP-L was missing two of the three conserved residues at the amidase catalytic sites. However, SiPGRP-L retained complete Zn^2+^ binding sites. These findings suggest that only SiPGRP-S1 may possess amidase activity within the SiPGRP gene family (Figure 4).

### 2.5. Protein Structure Prediction and Docking with PGN Ligand

To gain a deeper understanding of the relationship between the SiPGRP gene family, we utilized AlphaFold v2.1.0 (AF2) to construct three-dimensional models of the corresponding PGRP structural domains for the SiPGRP gene family and DmPGRP-SC2 (Figure 5A–E). Notably, all four PGRPs exhibit over 90% of their amino acid residues situated in the most favored regions, indicating the successful attainment of high-quality 3D structures for these proteins. These results demonstrate that these molecules possess analogous architectures, each comprising four α-helices (α1, α2, α3, α4, α5) and five β-strands (β1, β2, β3, β4, β5). Furthermore, all structures contain a disulfide bond between both β1 and α2, which enhances the structural integrity of the peptide chain. The root-mean-square deviation (RMSD) derived from the values superposition of SiPGRP-S1 and DmPGRP-SC2 is 1.08 Å (Figure 5F). Additionally, the catalytic centers of SiPGRP-S1 and DmPGRP-SC2 were superimposed, revealing a similar overall topology between the two structures.

Docking studies were conducted using Autodock Vina (v1.1.2) to investigate the capacity of SiPGRPs to recognize PGNs. As illustrated, we employed three ligands representing DAP-type PGN (TCT), LYS-type PGN (MTP), and laminarihexaose to interact with the amino acids of SiPGRP-S1 (Figure 6). The binding energies for these ligands were −35,145.6, −27,196, and −33,890.4 J/mol, respectively, indicating strong interactions. Specifically, TCT (HIS-59) and laminarihexaose (HIS-82) could form salt bridges with SiPGRP-S1. Additionally, hydrophobic interactions were observed between TCT and SiPGRP-S1 (TRP-88). Consequently, TCT exhibits a stronger interaction with SiPGRP-S1 compared to the other ligands. All ligands were reliably docked into the PGN-binding pocket of SiPGRP-S2, SiPGRP-S3, and SiPGRP-L, with binding energies exceeding −27,196 J/mol.

### 2.6. Expression Pattern Analysis of SiPGRPs

To investigate the expression relationship among *SiPGRPs* in healthy *S. invicta*, we profiled their expression across various life stages/castes (eggs, larvae, pupae, major, medium, minor, male, and female), as well as different tissues (head, thorax, and abdomen), and immune tissues (fat body, midgut, and hemocyte) using RT-qPCR. The mRNA level of *SiPGRP*-*S1* exhibited significantly higher expressions in major worker ants, while *SiPGRP*-*S2*, *SiPGRP*-*S3*, and *SiPGRP*-*L* showed comparatively reduced expression across all worker castes (Figure 7A), suggesting a potentially important role for *SiPGRP*-*S1* in *S. invicta.* Notably, *SiPGRP*-*S1* expression was significantly lower in immature stages (eggs, larvae, and pupae) and in adults (male and female) (Figure 7A). In contrast, a relatively high abundance of *SiPGRP*-*S2* and *SiPGRP*-*L* was observed during these periods (Figure 7A), indicating a complex regulatory interplay among *PGRP* genes in *S. invicta*. Additionally, *SiPGRP*-*L* exhibited the highest expression level in major worker ants. Furthermore, tissue-specific expression profiling revealed that *SiPGRPs* were abundantly expressed in abdominal tissues compared to the head and thorax (Figure 7B), with a predominant expression in fat body tissues relative to midgut and hemocyte tissues (Figure 7C).

### 2.7. Bacterial and Fungal Challenges Induced the Expression of PGRPs in S. invicta

To further investigate the induction of *SiPGRPs* by bacteria and fungi, we utilized Gram-negative bacteria, Gram-positive bacteria, and fungi to infest *S. invicta*, measuring the expression levels of *SiPGRPs* (Figure 8). Following infestation with fungi (*I. fumosorosea, M. anisopliae,* and *B. bassiana*), *SiPGRP*-*S1* and *SiPGRP*-*L* were significantly up-regulated, in contrast to the response observed with bacterial infestation. Notably, *SiPGRP*-*S2* exhibited a significant increase in expression following infestation with Gram-positive bacteria. Additionally, *SiPGRP*-*S3* was up-regulated after infestation with Gram-negative bacteria; however, its expression level was comparatively lower than that of the other three *SiPGRPs*.

We further assessed the expression levels of major and minor *S. invicta* following infection with *M. anisopliae* (Figure 9). The expression levels of *SiPGRPs* increased over time, peaking at 72 h. Notably, the expression of *SiPGRP*-*S1* was the most significantly up-regulated, reaching approximately twice the levels of the other *SiPGRPs*. Furthermore, the expression levels of *SiPGRPs* in the major group were significantly higher than those in the minor group after infestation by *M. anisopliae*.

### 2.8. Antimicrobial Activity of Peptides from the Hemolymph of S. invicta

The agar diffusion method was utilized to assess the bacteriostatic effect of AMPs in the hemolymph of major fire ants infected with *M. anisopliae*. Our observations indicated the growth of the indicator fungus *M. anisopliae* on the medium, and the colony count results demonstrated that AMPs derived from hemolymph of *M. anisopliae*-infected major fire ants exhibited significantly greater antifungal activity against *M. anisopliae* in vitro compared to the uninfected control (Figure 10A,B,E). Additionally, we noted the growth of indicator bacteria on the medium, revealing that the AMPs present in the hemolymph of *M. anisopliae*-infected major fire ants had a modest inhibitory effect on *S. aureus (*Figure 10D,G). At 36 h, 48 h, and 72 h post-infection, the diameter of the antibacterial zone formed by AMPs in the *M. anisopliae*-treated group was slightly larger than that observed in the control group (Tween-80-infected major fire ants). These results indicate that *M. anisoplia*-induced AMPs exert some inhibitory effects on *S. aureus* (Figure 10D,G). Notably, a pronounced antibacterial effect was observed against *E. coli* (Figure 10C,F). At 24 h, 36 h, 48 h, and 72 h following the infection of major fire ants with *M. anisopliae*, the diameter of the antibacterial zone formed by AMPs in the *M. anisopliae*-treated group was significantly larger than that in the control group (Tween-80-infected major fire ants) (Figure 10C,F), indicating statistical significance. This finding suggests that *M. anisopliae* infection in major fire ants elicits immune responses in *S. invicta*, leading to an increased synthesis of AMPs with potential antibacterial effects.

## 3. Discussion

The primary recognition receptors of pathogens are peptidoglycan recognition proteins, which are essential for environmental adaptation and resistance to a range of microbial diseases [31]. Insect PGRPs can be categorized based on their molecular size and structure as long-form PGRPs (PGRP-L) and short extracellular PGRPs (PGRP-S) [32]. For the first time, the PGRP gene family in *S. invicta* was found by this investigation. The number of PGRPs in Hymenoptera has decreased dramatically in comparison to the many *PGRP* genes found in the well-researched *D. melanogaster* [13] and *B. mori* [12]. There are only four known *PGRP* genes in *S. invicta*. Because they are social insects, ants have social immunity and will groom themselves when they come into contact with diseases. One unique kind of defense is social immunity. Social immunity is a special defense system that, in part, lessens the likelihood that viruses will infect hosts. Insects will also develop more robust immune systems in order to protect themselves under challenging environmental circumstances. In Hymenoptera, the number of *PGRPs* possessed by social insects is generally low, with most having only three (Appendix A). The fact that there are so many *SiPGRPs* might further indicate that *S. invicta* inhabits a very basic environment.

We conducted a thorough bioinformatics investigation of *S. invicta PGRP* genes. We made corrections to the *S. invicta* genome assembly problems and repositioned the *SiPGRPs* to their right locations on the chromosomes. The three kinds of *PGRP* genes found in *S. invicta* are *PGRP*-*SA*, *PGRP*-*SC*, and *PGRP*-*LC*, according to the phylogenetic tree. LYS-type PGN may be recognized by PGRP-SA of *D. melanogaster*, the zinc-binding site is mutated [33]. The *D. melanogaster* PGRP-SC has amidase activity and binds preferentially with DAP-type PGN [34].

The PGRP-LC of *D. melanogaster* is crucial for full activation of the IMD pathway [35]. In most PGRPs exhibiting with amidase activity, five key amidase active sites are characterized by the sequence of H-Y-H-T-C [36]. Through multiple sequence alignments, it was determined that only SiPGRP-S1 possesses complete amidase catalytic sites and zinc-binding sites. This indicates that SiPGRP-S1 is instrumental in recognizing peptidoglycan and activating the IMD signaling pathway in *S. invicta.*

Currently, the presence of antibacterial activity in PGRPs from various insect species has been confirmed. For instance, the PGRP-LB from *Rhynchophorus ferrugineus* and PGRP-S5 from *Bombyx mori* exhibit the ability to inhibit bacterial growth [37,38]. We predicted the tertiary structure of SiPGRPs, which share similarities in their structural composition. The predicted tertiary structures of DmPGRP-SC2 and SiPGRP-S1 show a high degree of similarity, suggesting potential functional parallels. Notably, DmPGRP-SC2 in *Drosophila* lacks bactericidal activity; however, it inhibits the activation of the IMD signaling pathway by cleaving PGN into inactive peptides and amino acids [33]. Furthermore, the molecular docking results indicate that SiPGRPs can effectively bind to three PGNs (Appendix A). The binding affinity of SiPGRP-S1 for TCT is particularly high, aligning with previously reported findings [16]. Key residues, TYR-93 and THR-173, which correspond to amidase catalytic sites, play a crucial role in the binding process. SiPGRP-LC demonstrates a strong binding capability to TCT. SiPGRP-S1 recognizes TCT and cleaves TCT, thereby inhibiting the activation of the IMD pathway by SiPGRP-LC. This mechanism serves to prevent the over-activation of the IMD pathway due to microbial spontaneity.

To investigate the functional diversity of *SiPGRPs* in response to peptidoglycan, we first analyzed the expression profiles of *SiPGRPs* induced by various bacteria and fungi. Previous studies have shown that short *PGRPs* in insects are predominantly expressed in fat bodies [8]. For instance, *BmPGRP*-*S* is highly expressed in the fat body in *B. mori* [39], while *PGRP*-*1* exhibits significantly higher expression levels than hemocytes in *Ostrinia furnacali* [40]. In contrast, long *PGRPs* are predominantly expressed in hemocytes [41,42]. Our evaluation of *SiPGRP* expression in immune tissues revealed significant up-regulation in the fat body following *M. anisopliae* infection. Notably, SiPGRP-S2 and SiPGRP-S3 are closely related to BmPGRP-S1 and BmPGRP-S2 in the phylogenetic tree. BmPGRP-S1 is known to bind to the PGN of Gram-positive bacteria and activate the prophenoloxidase (PPO) cascade [11]. Meanwhile, BmPGRP-S2 is associated with IMD pathway activation and shares a highly similar structure with PGRP-SA [43]. The expression profile aligned with this, as the *SiPGRP*-*S2* is significantly up-regulated following Gram-positive infection, whereas Gram-negative bacteria infestations led to a notable increase in *SiPGRP*-*S3* expression. Interestingly, infestation of *S. invicta* with *M. anisopliae* and *B. bassiana* resulted in significant increases in *SiPGRP*-*S1* and *SiPGRP*-*L* expression. However, while *SiPGRP*-*S1* and *SiPGRP*-*L* are implicated in regulating the activation of the IMD pathway, it remains unverified whether their activation is directly triggered by fungal-derived compounds. A more plausible explanation is that the microbiota imbalances caused by fungal infections, along with bacteria proliferation, lead to the activation of the IMD pathway.

As a eusocial insect, *S. invicta* exhibits a distinct mechanism of social division of labor [44]. In this species, the division of labor among worker ants is closely linked to their developmental age [45]. Minor workers primarily remain within the nest, tending to the queens and their offspring. As they mature, these workers transition to foraging roles outside the nest, at which point they are referred to as major workers [46]. Following the infection of both major and minor workers with *M. anisopliae,* we observed a progressive increase in the expression levels of *SiPGRPs*, peaking at 72 h post-infection. Notably, at 24 h, the expression level of *SiPGRPs* in major workers significantly increased, distinguishing them from minors. Insects can acquire bacterial symbionts through either vertical transmission or environmental exposure [47]. The greater exposure of major workers to the external environment results in a more complex microbial community structure, thereby enhancing their immune defense response against pathogen infections. Our evaluation of antibacterial activity revealed that major ants activated the IMD and Toll signaling pathways upon infection with *M. anisopliae*, leading to the production of antimicrobial peptides (AMPs) that effectively inhibited *S. aureus* and *E. coli*. Interestingly, the expression of SiPGRPs was suppressed at 24 h and 36 h post-infection but was significantly up-regulated at 48 h. Previous studies have indicated that *M. anisopliae* can produce secondary metabolites, such as destruxins, which may inhibit the insect immune response [48]. Consequently, this increases the likelihood of pathogen invasion, disrupting the homeostasis of the bacterial community structure. The pathogen stimulates the expression of SiPGRP-S1 and SiPGRP-L during competition, thereby activating a cascade of immune responses.

We identified and characterized four *PGRP* genes from *S. invicta.* Bioinformatics analyses and immune response assessments demonstrated that all four SiPGRP are involved in the immune defense against *M. anisopliae* and possess the capability to bind to Gram-positive bacteria, Gram-negative bacteria, and fungi. Notably, SiPGRP-S1 plays a significant role in the immune response process. Collectively, the four SiPGRPs regulate the expression of AMPs to defend against pathogen invasion. Our study enhances the understanding of the role of PGRPs in the innate defense immune defense mechanism of *S. invicta*, and aims to improve the efficacy of microbial insecticides by integrating knowledge of their regulatory functions.

## 4. Materials and Methods

### 4.1. Insects and Microorganisms

*S. invicta* specimens were collected from Huangpu District in Guangzhou, Guangdong Province, China, and were maintained in a 20 L plastic container internally coated with Fluon^®^ (AGC Chemicals Trading, Shanghai, China) to prevent escape. The environmental conditions were kept at 26 °C, with a relative humidity (RH) of 70% and a photoperiod of 12 h light and 12 h dark. The ants were provided with a diet consisting of minced mealworms *Tenebrio molitor,* or honey water (25%) soaked cotton balls. Since worker ants exhibit caste polymorphism, we classified them into three castes “Major” for those with a head width greater than 1.000 mm, “Medium” for those with a head width between 0.595 mm and 1.000 mm, and “Minor” for those with a head width less than 0.595 mm. Head width measurements were obtained using a digital micrometer (FB70252; Thermo Fisher Scientific, Waltham, MA, USA) under a stereomicroscope (SMZ-1500; Nikon, Tokyo, Japan).

*Escherichia coli* DH5α, *Staphylococcus aureus*, and *Serratia marcescens* were maintained at the State Key Laboratory of Green Pesticide, Guangzhou, Guangdong province, P. R. China. The highly pathogenic *B. thuringiensis* HD-73 strain was provided by Professor Zhang Jie of the Institute of Plant Protection, Chinese Academy of Agricultural Sciences. Gram-positive bacteria (*S. aureus* and *B. thuringiensis)* and Gram-negative bacteria (*S. marcescens* and *E. coli* DH5α) were cultivated in LB broth at 37 °C until reaching mid-log bacteria (5 × 10^7^ CFU/mL). The fungal entomopathogen *M. anisopliae* (strain MaqS1902), generously supplied by Qiongbo Hu from South China Agricultural University, Guangzhou, China, along with the *Isaria fumosoroseus* IfB01 strain (CCTCC M 2012400) and *Beauveria bassiana* Bb-01, was cultured on potato dextrose agar (PDA) for two weeks in complete darkness at 26 ± 1 °C and 75–80% RH [28]. The conidia were harvested into 0.05% Tween-80 (P1754, Sigma Aldrich, San Louis, MO, USA), and the desired concentrations of 5 × 10^7^ spores/mL were prepared, as described in our previous studies.

### 4.2. Identification of PGRP Genes in S. invicta

To identify members of the *PGRP* gene family in *S. invicta*, we retrieved the published protein sequences (GCF_016802725.1) from the National Center for Biotechnology Information (NCBI) protein database (https://www.ncbi.nlm.nih.gov/), accessed on 19 April 2024. Utilizing the *PGRP* hidden Markov model (PF01510) from the Pfam database (http://pfam-legacy.xfam.org/), we searched for the candidate *PGRP* protein sequences within the *S. invicta* protein dataset using HMMER 3.3 software, applying a cutoff E-value of <10^−5^. The search was conducted on 19 April 2024.

To refine the chromosomal positioning of *SiPGRPs*, we downloaded the genome file of *S. invicta* (GCA_018691235.1) from the NCBI database to serve as a reference genome. We assembled the source files obtained from our transcriptome sequencing using Bowtie2 (v2.3.5.1) and Samtools (v1.7). The approximate locations of *SiPGRPs* on the reference genome were determined through BLAST analysis. Visualization of the genome file and BAM file was conducted using IGV-GSAman (v0.8.4), while Softberry (http://www.softberry.com/) was employed to predict the structural features at the corresponding chromosome positions, accessed on 21 April 2024. Finally, conserved domains were identified on NCBI, allowing us to compile the members of the *PGRP* gene family in *S. invicta.*

The amino acid sequences obtained were analyzed for physicochemical properties using the EXPASY (Expert Protein Analyst System) platform (https://web.expasy.org/protparam/), accessed on 28 April 2024. Signal peptides were predicted utilizing the Signal P 5.0 server (https://services.healthtech.dtu.dk/services/SignalP-5.0/), accessed on 28 April 2024. The locations of the genes on the chromosomes were mapped using TBtools-II (v2.096). Additionally, the structural map of the gene was created using IBS (v1.0).

### 4.3. Collinearity Analysis of SiPGRPs

We obtained the published protein sequences of *D. melanogaster*, *B. mori*, *M. pharaonis,* and *A. cerana* from the NCBI database, accessed on 1 May 2024. Subsequently, we identified the members of their *PGRP* gene family. Homologous genes were searched using OrthoFinder (v2.5.5) and visualized using JCVI (v1.4.16).

### 4.4. Phylogenetic Analysis and Sequence Alignment of SiPGRPs

Representative PGRP protein sequences from 21 species were retrieved from the NCBI database (Appendix A). The amino acid sequences were aligned using the BLAST (v2.14.0) algorithm with default parameters. A phylogenetic tree was constructed employing the maximum likelihood method in IQ-TREE software (v2.2.2.9), utilizing 1000 bootstrap repetitions for robustness. The resulting tree was visualized using iTOL v6.9.1 (https://itol.embl.de/), accessed on 5 May 2024. Additionally, multiple sequence alignment of partial sequences was conducted using ClustalX2 (v2.1), and the alignment results were refined for clarity using GeneDoc (v2.7).

### 4.5. Structure Prediction and Molecular Docking

The three-dimensional (3D) structure of SiPGRPs was predicted using AlphaFold2 (v2.3.0). Molecular structure superposition was conducted with PyMOL (v2.6.0). The quality of the predicted model was assessed using SWISS-MODEL (https://swissmodel.expasy.org/), accessed on 10 May 2024. Corresponding grid boxes for various receptors were established for molecular docking using Autodock vina (v1.1.2). The optimal results were visualized using PyMOL (v2.6.0). TCT (PDB: 2F2L), MTP (PDB: 1TWQ), and laminarihexaose (PDB: 1W9W) represent Dap-type PGN, LYS-type PGN, and fungal PGN, respectively. Structure files were retrieved from the Protein Data Bank database (PDB).

### 4.6. Real-Time Quantitative PCR (RT-qPCR)

Total RNA was extracted from various tissues (head, thorax, abdomen, hemocytes, midgut, and fat body) and life stages/castes (eggs, larvae, pupae, major, medium, minor, male, and female) using TRIzol (Omega, Norcross, GA, USA). First-strand cDNA synthesis was conducted using the Color Reverse Transcription Kit (EZB, #A0010CGQ), adhering to the manufacturer’s guidelines. RT-qPCR was performed using 200 ng of cDNA and TB Green Premix ExTaq (TAKARA, #RR520A) utilizing three technical replicates on a CFX96^TM^ real-time PCR detection system (Bio-Rad, Hercules, CA, USA). The PCR program consisted of an initial step at 95 °C for 30 s, followed by 40 cycles of 95 °C for 5 s, and 55 °C for 10 s, a dissociation curve was generated from 65 to 95 °C to confirm product purity [49]. Data analysis was conducted using the 2^−ΔΔCT^ method [50], and results were presented as means from 3 biological replicates with standard deviation (Mean ± SD). The primers utilized in this study are detailed in Appendix A.

### 4.7. Microbial Induction and Time Course Expression of SiPGRPs

To analyze the induction of *SiPGRPs* in *S. invicta*, heat-killed Gram-positive bacteria *S. aureus*, *B. thuringiensis* (2.0 × 10^7^ CFU per larvae), and Gram-negative bacteria *S. marcescens*, *E. coli* DH5a (2.0 × 10^7^ CFU per larvae) were administered to *S. invicta* major workers, separately, while phosphate-buffered saline (PBS) was injected into the control group.

For the fungal infection experiment, three groups were selected from *S. invicta* major workers (30/replicate × 3 replicates/assay) and were exposed to suspensions of *M. anisopliae* (5 × 10^7^ CFU/mL), *I. fumosorosea* IfB01 (5 × 10^7^ CFU/mL), and *B. bassiana* Bb-01suspensions (5 × 10^7^ CFU/mL) for 4–5 s. Following exposure, the workers were placed onto filter paper to absorb excess liquid from their body surfaces, as described in our previous studies. The control group was treated with aqueous 0.05% Tween-80 (Sigma Aldrich, P1754). Whole bodies were collected 24 h post-infection for RNA isolation, first-strand cDNA synthesis, both performed as outlined above. The experiment was performed with three biological replicates, and RT-qPCR was performed as outlined above.

To further analyze the induction of *SiPGRPs* by *M. anisopliae* in both major and minor fire ants, suspensions of *M. anisopliae* (5 × 10^7^ CFU/mL) were applied for 4–5 s, after which the ants were placed on filter paper to absorb any excess liquid from their body surfaces, as described in our previous studies [51]. The control group was treated with an aqueous solution of 0.05% Tween 80 (Sigma Aldrich, P1754). Whole bodies were collected at 0, 6, 12, 24, 36, 48, and 72 h post-infection. Samples from 10 larvae were pooled to create a biological sample for RNA isolation and first-strand cDNA synthesis, both performed as outlined above. The experiment was performed with three biological replicates, and RT-qPCR was performed as outlined above.

### 4.8. Antibacterial Activity

To assess the bacteriostatic activity of AMPs, hemolymph was collected from 50 major fire ants infected with *M. anisopliae* at 0, 6, 12, 24, 36, 48, and 72 h post-infection (hpi), respectively. Hemolymph was heat-treated at 100 °C and centrifuged at 5000× *g* for 10 min at 4 °C, after which, the supernatant was stored at −80 °C for later use [52]. The agarose medium was prepared by melting and then cooling it to approximately 50 °C. *E. coli* DH5α, with an optical density at 600 nm (OD_600_) of 0.3, OD_600_ = 0.3, was added to the agarose medium at a dilution of 1:1000, and the mixture was poured into petri dishes. Once the medium had completely solidified, it was perforated by creating wells approximately 3 mm in diameter. Subsequently, 20 μL of the hemolymph supernatant was added to each well, allowing for complete diffusion within 1 h at room temperature. The petri dishes were then incubated overnight at 37 °C, and the diameter of the inhibition zones was observed. The experiment was conducted in triplicate.

For the antifungal activity assays, *M. anisopliae* was transferred to potato dextrose agar (PDA) plates and cultured for two weeks at 25 °C, and 0.05% Tween-80 solution was added into the plate to create a spore suspension. Ten microliters of the heat-treated hemolymph supernatant sample were mixed with 80 μL of active spore suspension (2.4 × 10^6^ CFU/mL), and the resulting mixture was evenly spread on a PDA plate supplemented with appropriate antibiotics, including Amp^+^ (100 μg/mL) and Kan^+^ (50 μg/mL). The plates were subsequently incubated for 40 h, after which colony-forming units (CFUs) were counted. The experiment was conducted in triplicate.

## Figures and Tables

**Figure 1 ijms-25-12289-f001:**
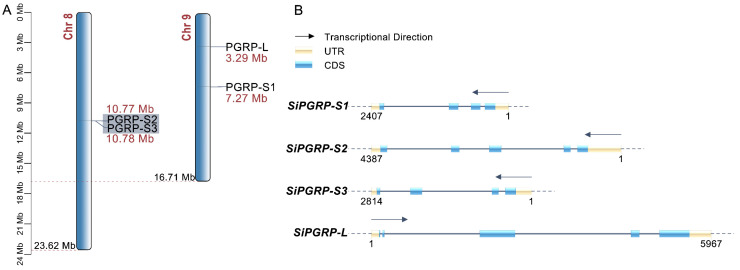
(**A**) Chromosome mapping of members of the *PGRP* gene family in *S. invicta.* The number of genes corresponding to each chromosome is indicated on the left. (**B**) Schematic presentation of the protein structure of *SiPGRPs*, with the start and end sites labeled below these genes.

**Figure 2 ijms-25-12289-f002:**
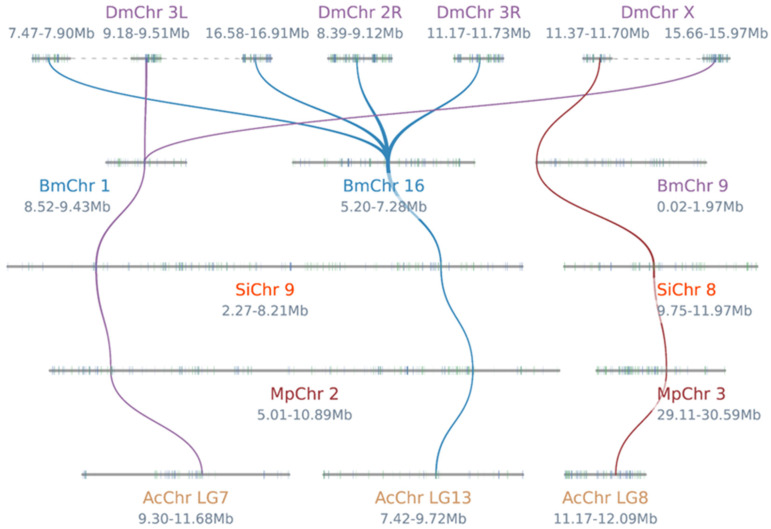
The collinearity relationships of the *PGRP* gene pairs from the five species (*D. melanogaster*, *B. mori*, *S. invicta, M. pharaonis*, and *A. cerana*). The corresponding gene start and end sites are marked below the chromosome name. The purple lines indicate gene pairs homologous to *SiPGRP*-*L*. The blue lines delineate gene pairs homologous to *SiPGRP*-*S1*. The gene pairs homologous to *SiPGRP*-*S2* and *SiPGRP*-*S3* are marked with red lines.

**Figure 3 ijms-25-12289-f003:**
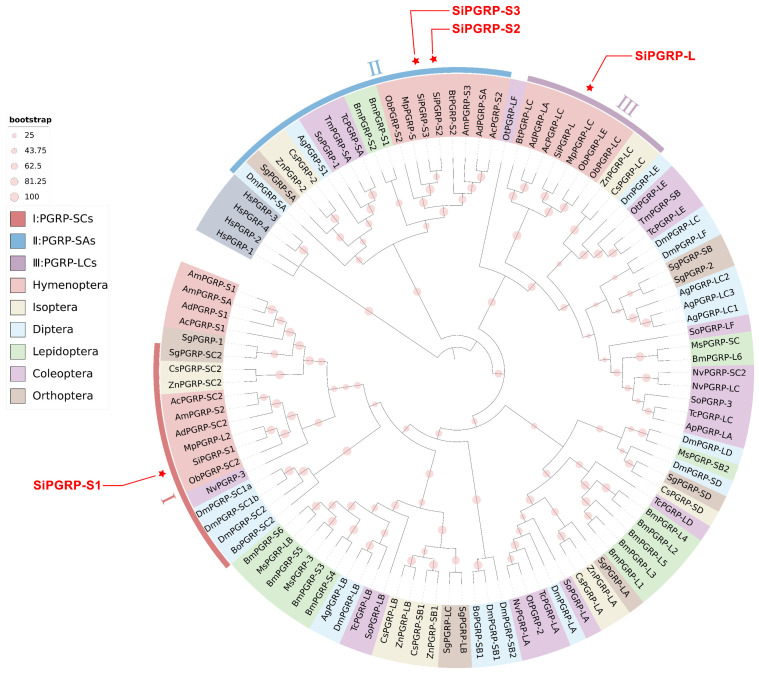
Phylogenetic relationships of PGRPs were analyzed by constructing a maximum likelihood tree based on the amino acid sequences from 22 species across six orders (Appendix A). The optimal model employed for this analysis was Q. pfam + R5. PGRPs from different orders were distinguished using various colored backgrounds. The SiPGRPs were depicted as pentagrams and were categorized into three distinct groups, each represented by a different color in the outer ring.

**Figure 4 ijms-25-12289-f004:**
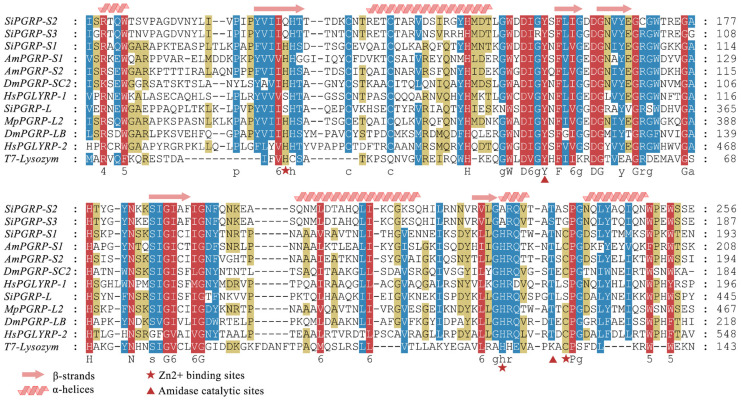
The sequence alignment of SiPGRPs with PGRPs from other insects is presented. Key residues critical for T7 lysozyme activity are marked with pentagrams and triangles. The α-helices and β-strands are indicated by spiral patterns and arrows at the top of the diagram. Identical residues (100%) are highlighted in red, while similarities of 75% and 50% are shown in blue and yellow, respectively. The GenBank accession numbers for the sequences are as follows: AmPGRP-S1 (XP_001121036.2), AmPGRP-S2 (NP_001157188.1), DmPGRP-SC2 (NP_610410.1), DmPGRP-LB (NP_001247052.1), MpPGRP-L2 (XP_012534615.2), HsPGLYRP-1 (NP_005082.1), HsPGLYRP-2 (NP_001350475.1), and T7-Lysozym (NP_041973.1).

**Figure 5 ijms-25-12289-f005:**
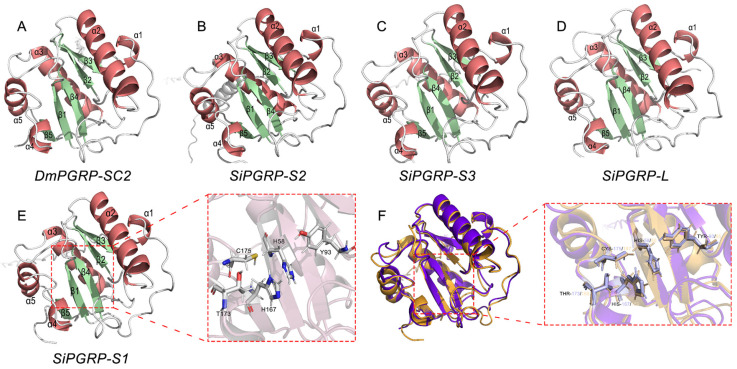
3D structures of DmPGRP-SC2 and SiPGRPs. Full structure: (**A**) DmPGRP-SC2, (**B**) SiPGRP-S2, (**C**) SiPGRP-S3, (**D**) SiPGRP-L, (**E**) SiPGRP-S1. α-helices are represented by ribbons of red and the β-strands are green. (**F**) SiPGRP-S1 (purple) and DmPGRP-SC2 (yellow) were superposed.

**Figure 6 ijms-25-12289-f006:**
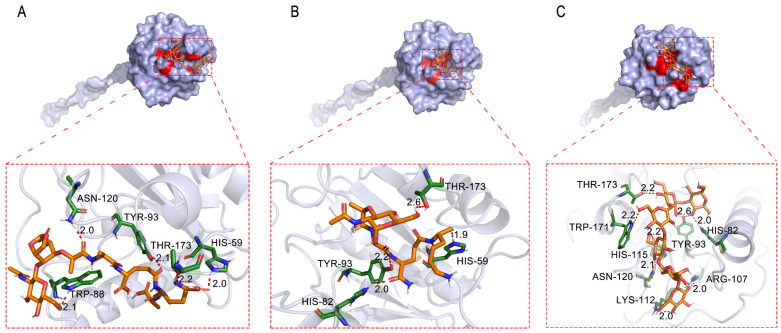
The docking view illustrates the binding of SiPGRP-S1 to various ligands. (**A**) TCT (tracheal cytotoxin), (**B**) MTP (muramyl tripeptide), and (**C**) laminarihexaose. Hydrogen bonds are represented by red dashed lines, while the ligands are depicted as orange sticks. The interacting residues of the receptor are shown as green sticks. The residues that form hydrogen bonds with the three ligands are as follows: (**A**) HIS-59, TRP-88, TYR-93, ASN-120, THR-173, (**B**) HIS-59, HIS-82, TYR-93, THR-173, (**C**) HIS-82, TYR-93, ARG-107, LYS-112, HIS-115, ASN-120, TRP-171, THR-173.

**Figure 7 ijms-25-12289-f007:**
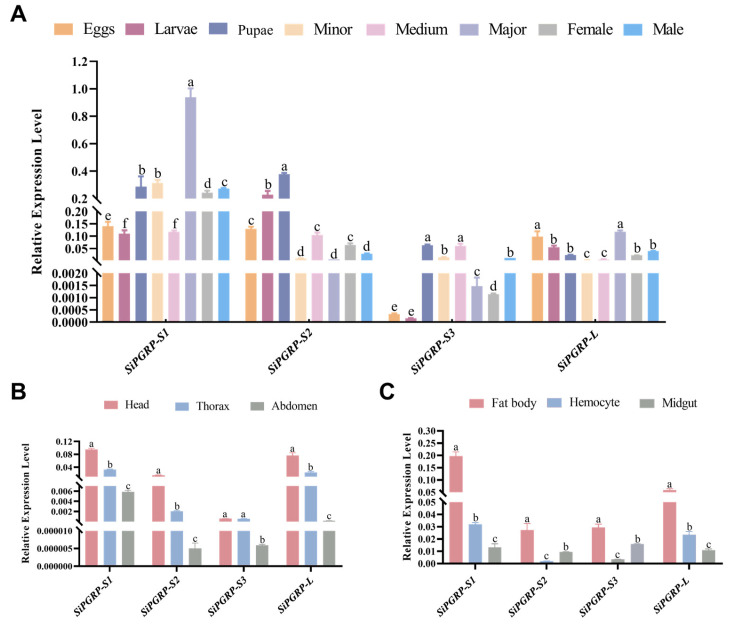
The relative expression level of *SiPGRPs* in *S. invicta*. Expression levels at (**A**) various life stages/castes, (**B**) body parts, and (**C**) immune tissues. Different letters above bars indicate significant difference from one another (*p* < 0.05; ANOVA followed by a post hoc Tukey’s honestly significant difference [HSD] test).

**Figure 8 ijms-25-12289-f008:**
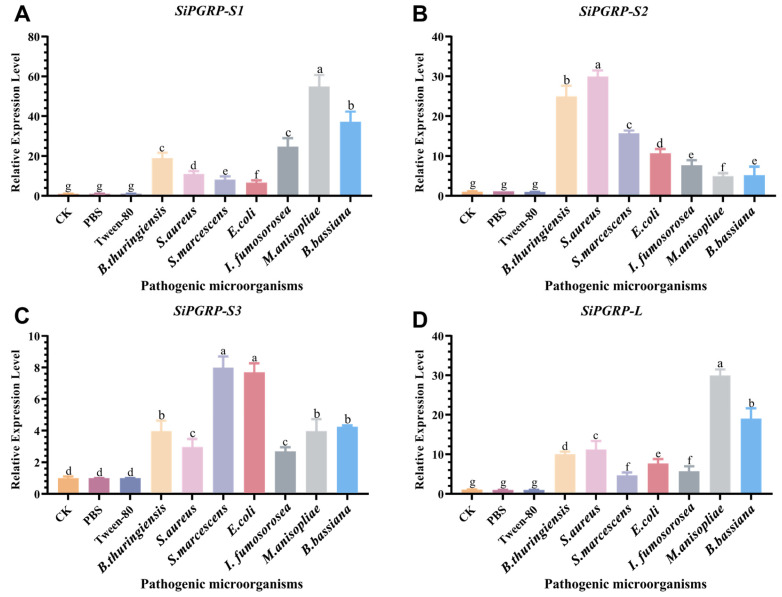
The RT-qPCR analysis of *SiPGPRs* expression levle of in reponse to various pathogen in *S. invicta*. (**A**) *SiPGRP-S1*, (**B**) *SiPGRP-S2*, (**C**) *SiPGRP-S3* and (**D**) *SiPGRP-L* after infection by various bacteria and fungi. CK, control check; PBS, phosphate buffer saline; *B. thuringiensis*, *Bacillus thuringiensis*; *S. aureus*, *Staphylococcus aureus*; *S. marcescens*, *Serratia marcescens*; *E. coil*, *Escherichia coli*; *I. fumosorosea*, *Isaria fumosorosea*; *M. anisopliae*, *Metarhizium anisopliae*; *B. bassiana*, *Beauveria bassiana*. Different letters above bars indicate significant difference from control (*p* < 0.05; ANOVA followed by a post hoc Tukey’s honestly significant difference [HSD] test).

**Figure 9 ijms-25-12289-f009:**
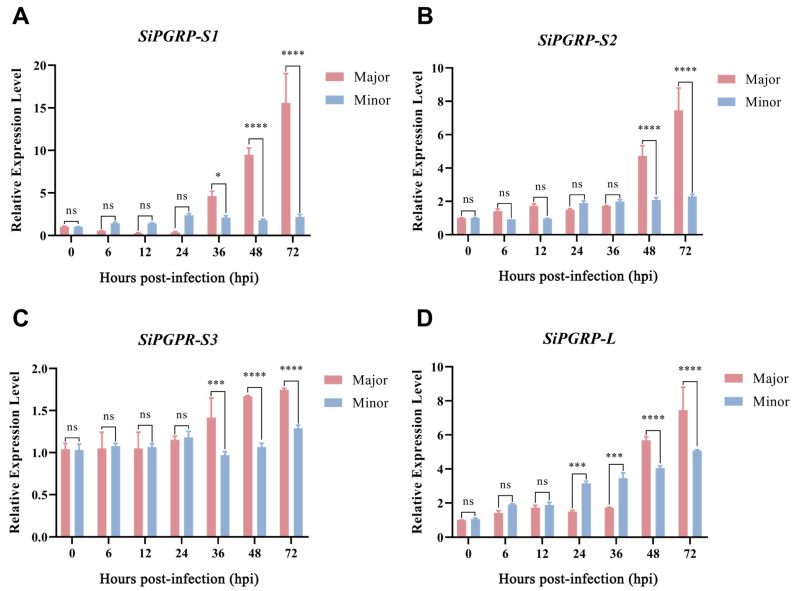
The RT-qPCR analysis of *SiPGPRs* expression levle of in reponse to various pathogen in *S. invicta*. (**A**) *SiPGRP-S1*, (**B**) *SiPGRP-S2*, (**C**) *SiPGRP-S3* and (**D**) *SiPGRP-L* after infection by various bacteria and fungi. We used a t-test and one-way analysis of variance (ANOVA) to analyze the data. The “*”, “***”, and “****” represent significant differences at *p* < 0.05, *p* < 0.001, and *p* < 0.0001, respectively. The “ns” represents non-significant differences.

**Figure 10 ijms-25-12289-f010:**
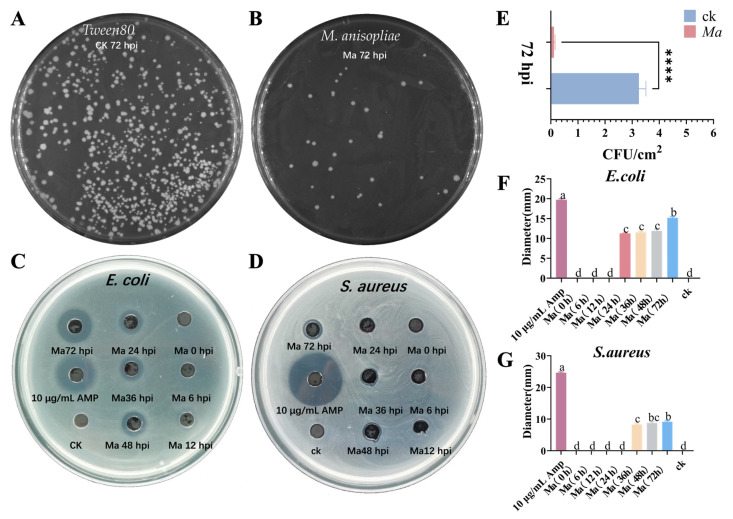
Infection with *M. anisopliae* activated the antibacterial properties of hemolymph in *S. invicta*. Major fire ants were infected with *M. anisopliae*, while a control group was maintained using 0.05%Tween80 (CK). Hemolymph was collected at 0, 6, 12, 24, 36, 48, and 72 hpi to assess antibacterial activity. (**A**,**B**) Antifungal assays were conducted using hemolymph from *M. anisopliae*-infected major fire ants at 72 h. (**C**,**D**) Representative inhibition zone assays were performed using hemolymph from *M. anisopliae*-infected major fire ants against *E. coli* and *S. aureus*. (**E**) Mean ± SEM of the colony count of *M. anisopliae* for three replicates (The “****” represent significant differences at *p* < 0.0001). Mean diameter (mm) ± SEM of three inhibition zone assays of *E. coli* (**F**) and *S. aureus* (**G**). Different letters above bars indicate significant difference from one another (*p* < 0.05; ANOVA followed by a post hoc Tukey’s honestly significant difference [HSD] test).

**Table 1 ijms-25-12289-t001:** Physicochemical properties of *PGRP* genes identified in *S. invicta*.

Name	Gene Length (bp)	CDS Length (bp)	AA	Domain	Molecular Weight (Da)	Instability Index	PI	GRAVY
SiPGRP-S1	2406	585	194	S-PGRP-Ami_2	21,586.49	14.79	9.02	−0.363
SiPGRP-S2	4386	774	286	TM-PGRP-Ami_2	32,908.62	34.24	9.62	−0.499
SiPGRP-S3	2813	567	236	S-PGRP-Ami_2	26,704.49	27.1	9.17	−0.337
SiPGRP-L	5966	1341	446	TM-PGRP-Ami_2	49,281.26	38.05	5.72	−0.371

## Data Availability

No data were used for the research described in the article.

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
