# Peer review of "Comparative Analysis of PGRP Family in Polymorphic Worker Castes of Solenopsis invicta"

_ijms, 2024, doi:10.3390/ijms252212289_

Round 1

Reviewer 1 Report

Comments and Suggestions for Authors

This manuscript contains a number of minor typographical and grammatical errors (the latter being mainly use of singulars instead of plurals and vice versa) plus some missing words in places as well as some duplicated text, e.g. lines 393-394.  The text should therefore be carefully checked to eliminate these.

Please state the full version/meaning of all acronyms on first use, e.g. for PI and GRAVY, lines 103, 104, and Table 1.  There are others not mentioned here.

Please give the full scientific names of all species on first use.

The species name of Monomorium pharaonis is misspelled, lines 125, 130, 436, and key to Figure 2.

It would be really helpful if the life stages shown in the key to Figure 7A were aligned in the same sequence that they appear in the histograms.  The colours of all figures are so similar that many readers may have difficulty distinguishing one stage from another so better contrasting, and preferably, primary colours should be used rather than pastel shades.

Line 238: the reference to Figure 7 should be to Figure 8, where responses to infections are shown.

The key to Figure 8 does not include the abbreviations for the control tests.

Line 496: What temperature was the heat treatment?

The text size on Figures 1 and 3 should be increased to make it readable.

Author Response

Reviewer #1:

Author Reply: We sincerely thank you for your valuable feedback and constructive suggestions, which have significantly enhanced the quality of our manuscript. Each comment has been addressed individually in the response letter. We hope that these revisions have adequately resolved your concerns. We genuinely appreciate your insightful input.

Comments 1: This manuscript contains a number of minor typographical and grammatical errors (the latter being mainly use of singulars instead of plurals and vice versa) plus some missing words in places as well as some duplicated text, e.g. lines 393-394. The text should therefore be carefully checked to eliminate these.

Response 1: In response to your valuable suggestions, we have thoroughly revised the entire text to correct typographical and grammatical errors, including those noted in Lines 50, 99, 81, 424, and 486. We have also addressed the missing and repeated words throughout the manuscript, specifically in Lines 65, 96, 269, 395, 477, 483, 486, 491, and 513. Additionally, we have corrected instances of missing spaces in the text, as noted in Lines 91, 147, and 155.

Comments 2: Please state the full version/meaning of all acronyms on first use, e.g. for PI and GRAVY, lines 103, 104, and Table 1.  There are others not mentioned here.

Response 2: Thank you for your helpful advice. We have added the full versions for the acronyms 'PI' (isoelectric point) and 'GRAVY' (grand average of hydropathicity) on lines 103 and 104, respectively.

Comments 3: Please give the full scientific names of all species on first use.

Response 3: We appreciate your valuable input regarding the scientific names of all species.We have made the relevant changes as per your suggestion on line 124.

Comments 4: The species name of Monomorium pharaonis is misspelled, lines 125, 130, 436, and key to Figure 2.

Response 4: Thank you for your meticulous review and for identifying the misspelling of the species name, Monomorium pharaonis. We have corrected this error in the manuscript, specifically on lines 125, 130, 138, and 436.

Comments 5: It would be really helpful if the life stages shown in the key to Figure 7A were aligned in the same sequence that they appear in the histograms. The colours of all figures are so similar that many readers may have difficulty distinguishing one stage from another so better contrasting, and preferably, primary colours should be used rather than pastel shades.

Response 5: We have reorganized the order of the legends in Figure 7A in accordance with your recommendation. Additionally, the figures in the full article have been updated to feature more contrasting colors.

Comments 6: Line 238: the reference to Figure 7 should be to Figure 8, where responses to infections are shown.

Response 6: Thank you for your valuable feedback. We have made the recommended changes accordingly in Line 238.

Comments 7: The key to Figure 8 does not include the abbreviations for the control tests.

Response 7: We appreciate you bringing this to our attention. We have supplemented this information as needed in Lines 246-247.

Comments 8: Line 496: What temperature was the heat treatment?

Response 8: The heat treatment was conducted at a temperature of 100, and we have included this detail in the text at Line 496.

Comments 9: The text size on Figures 1 and 3 should be increased to make it readable.

Response 9: We sincerely appreciate your feedback regarding the clarity of our figures. We have appropriately increased the text size in Figures 1 and 3.

Reviewer 2 Report

Comments and Suggestions for Authors

This paper looks at four PGRPs in different tissues of worker ants of RIFA. It includes in silico analysis of the PGRPs, qPCR of different tissues and different castes, and immune challenges.

The research is thorough, covering many aspects of the topic. The figures are excellent. The discussion is supported by the results. I have no comments to make regarding the science of this paper.

Minor Comments
22-23: lowercase the tissue names
66-67 and 70: This is an international journal with international readership, so China-specific data is not interesting, though you can keep the citations.
93: missing space in "genome may"
Figure 1: Not essential, but can parts of this be edited to have larger font?
246-249 Italicize the species
290 space needed after (F)
300-301 "There are only four known PGRP genes." In S. invicta, or in the world? Specify.
306 "so many SiPGRPs" Is four a lot or a little? How many are in other insects?

Author Response

RESPONSE TO REVIEWERS

Reviewer #2:

This paper looks at four PGRPs in different tissues of worker ants of RIFA. It includes in silico analysis of the PGRPs, qPCR of different tissues and different castes, and immune challenges.
The research is thorough, covering many aspects of the topic. The figures are excellent. The discussion is supported by the results. I have no comments to make regarding the science of this paper.

Author Reply: We appreciate your expert review and constructive feedback. We have carefully considered each of your concerns and made significant revisions to the manuscript accordingly. In the revised version, we have included additional information to address the points raised and clarified certain aspects to enhance the overall clarity and depth of the manuscript. We hope these revisions adequately address your concerns. Thank you for your valuable input.

Comments 1: 22-23: lowercase the tissue names

Response 1: We appreciate your feedback. Based on your suggestion, we have reviewed and revised the full text, specifically addressing the tissue names in lowercase on Line 22, as well as Lines 458-459 and 216-218.

Comments 2: 66-67 and 70: This is an international journal with international readership, so China-specific data is not interesting, though you can keep the citations.

Response 2: In response to Comments 2: Lines 66-67 and 70, we acknowledge that this is an international journal with a diverse readership, and therefore, data specific to China may not be of broad interest. Consequently, we have revised this section of the manuscript to enhance its relevance to a wider audience. The specific data has been removed, including reference 20, as indicated in Lines 67-69.

Comments 3: 93: missing space in "genome may" Figure 1: Not essential, but can parts of this be edited to have larger font?

Response 3: Thank you for bringing this to our attention. We have thoroughly reviewed the entire manuscript for any missing spaces and have made the necessary revisions. Specifically, we addressed the issues identified in Line 91, Line 147, and Line 155.

Comments 4: 246-249 Italicize the species

Response 4: We have reviewed and standardized the use of italics in the manuscript in accordance with your suggestion. This includes the sections on lines 246-249, 289, and 291.

Comments 5: 290 space needed after (F)

Response 5: Thank you for your meticulous attention to detail in identifying this inconsistency. We have corrected the missing spaces in Line 290.

Comments 6: 300-301 "There are only four known PGRP genes." In S. invicta, or in the world? Specify.

Response 6: We appreciate your feedback. The phrase "four known PGRP genes" specifically refers to S. invicta, and we have clarified this in the manuscript on line 301.

Comments 7: 306 "so many SiPGRPs" Is four a lot or a little? How many are in other insects?

Response 7: Thank you for your insightful questions. We appreciate the opportunity to provide clarification. We have previously predicted all members of the PGRP gene family for social insects within the Hymenoptera order, specifically those whose genomes are assembled at the chromosome level. Most species possess only three PGRP gene family members. Among our predicted results, Ooceraea biroi is the only species with more members than Solenopsis invicta, having a total of six. Therefore, SiPGRP genes can be considered relatively abundant. We have made relevant additions to the manuscript, and Figure S2 has been included in the supplemental material, specifically on lines 307–308.
